

# Acute effect of combined exercise with aerobic and resistance exercises on executive function

Ying-Chu Chen[1], Ruei-Hong Li[1], Feng-Tzu Chen[2], Chih-Han Wu[3], Chung-Yu Chen[4], Che-Chien Chang[3] and Yu-Kai Chang[1]

[1] Department of Physical Education and Sport Sciences, National Taiwan Normal University, Taipei City, Taiwan
[2] Department of Sports Medicine, China Medical University, Tai-Chung City, Taiwan
[3] Office of Physical Education, National Central University, Taoyuan City, Taiwan
[4] University of Taipei, Department of Exercise and Health Sciences, Taipei City, Taiwan

Corresponding authors
Chung-Yu Chen,
fish0510@gmail.com
Che-Chien Chang,
watchtennis@yahoo.com.tw

## ABSTRACT

**Objective.** Recent studies indicate that acute exercise, whether aerobic exercise (AE) or resistance exercise (RE), improves cognitive function. However, the effects on cognitive function of combined exercise (CE), involving both AE and RE in an exercise session, remain unknown. The aim of this study was to investigate the effects of acute CE on cognitive function.

**Design.** Within-subject design with counterbalancing.

**Methods.** Fifteen healthy men with a sedentary lifestyle in the previous three months were recruited. The participants were assessed for muscular fitness after performing four upper body exercises for a 10-repetition maximum and underwent a submaximal aerobic fitness assessment for $\dot{V}O_{2peak}$ and corresponding workload (watts). They were then assigned to a CE, RE, or sitting control (SC) session in counterbalanced order and were assessed with the Stroop Color and Word Test (SCWT) after each session.

**Results.** Acute CE led to a significantly shorter response time compared to SC ($p < .05$) in the SCWT, wherein there were no significant differences between acute CE and RE ($p = 1.00$). Additionally, no significant differences in the accuracy rate were observed across the different sessions ($ps > .05$).

**Conclusion.** A single session of moderate-intensity CE improved response time in the SCWT, comparable to RE. CE shows promise for enhancing cognitive function, warranting further research on its benefits and other exercise modalities.

# INTRODUCTION

Acute exercise, also known as a single bout of exercise, is linked to a wide range of cognitive functions, including problem-solving (*Frith, Miller & Loprinzi, 2022*), verbal fluency (*Aguirre-Loaiza et al., 2019*), decision-making (*Lefferts et al., 2019*), attention, and executive functions (*Basso & Suzuki, 2017*; *Tsuk et al., 2019*). Recent meta-analytical reviews have further suggested that cognitive performance is improved, regardless of

whether it is measured during exercise or after the cessation of exercise (*Chang et al., 2012*; *Ishihara et al., 2021*; *Johnson et al., 2016*). The beneficial effects of acute exercise on cognitive function are believed to be due to the induction of the circulating brain-derived neurotrophic factor (*Ekblom et al., 2022*; *Hung et al., 2018*; *Tsai et al., 2021*), insulin-like growth factor 1 (*Arazi et al., 2021*; *De Alcantara Borba et al., 2020*; *Stein et al., 2018*), and neural plasticity (*Ben-Zeev, Shoenfeld & Hoffman, 2022*; *Walsh & Tschakovsky, 2018*). Despite the cognitive benefits of acute exercise, its effects on cognitive function are heterogenous, suggesting that they might be moderated (*Chang et al., 2012*; *Johnson et al., 2016*). For instance, while aerobic exercise (AE) leads to positive effects on cognitive function, anaerobic exercise leads to negative effects; interestingly, the combination of the two exercise types shows the largest positive effect (*Chang et al., 2012*), highlighting the importance of determining the exercise type in order to maximize the benefits for cognitive function.

Combined exercise (CE), also known as concurrent exercise (*Brito, Soares & Silva, 2019*), has been recognized as an effective and viable approach to enhance physical fitness (*Bouamra et al., 2022*; *Markov, Hauser & Chaabene, 2022*; *Moghadam et al., 2020*). CE involves the integration of AE and resistance exercise (RE) within a single exercise session, and its initial investigation dates back to Hickson's study in 1980 (*Hickson, 1980*). The main appeal of CE is that it not only meets the World Health Organization's guidelines on physical activity, which emphasize the importance of aerobic exercise and muscle-strengthening exercise (*Bull et al., 2020*; *WHO, 2020*), but also can be completed in a short period of time while having the same physiological adaptations compared to doing AE or RE separately (*Markov, Hauser & Chaabene, 2022*). Indeed, CE is recognized as a time-saving and beneficial strategy for improving body composition, strength, and hypertrophy and may have additional benefits compared to doing RE or AE separately.

Nevertheless, the acute effects of CE on cognitive function are still unknown. In fact, to the best of our knowledge, only two studies have investigated this topic. *Chang et al. (2017)* found that low-intensity total-body machine-based RE performed in a circuit-training mode combining low intensity aerobic exercise, such as walking on a treadmill, improves information processing speed and inhibition control of young college females. Interestingly, *Wen & Tsai (2020)* found that a single session of moderate-intensity body weight or dumbbell resistance exercise combined with moderate intensity aerobic dance exercise in a circuit-training form improves the neurophysiological inhibition control of obese women but does not improve behavioral performance parameters, such as accuracy rates and reaction times. In summary, investigations of the acute effects of CE on cognitive function are relatively limited, the findings of previous studies are inconsistent, and the mechanism underlying the enhancement of exercise-induced cognitive function remains unknown. Furthermore, whether different modes of CE, which combine traditional sequential movement resistance exercise, are beneficial for exercise-induced cognitive function remains to be clarified. Furthermore, acute RE has garnered recognition as a significant and promising exercise modality within the realm of exercise and cognitive function (*Huang et al., 2022*; *Wilke et al., 2019*); however, there are relatively few studies investigating the effects of RE on cognitive function compared to study targeted on AE

(*Chow et al., 2021*). Therefore, the present study specifically incorporates RE condition for comparison, aiming to address the limitations of previous studies in this area.

The aim of the present study was to investigate the immediate effects of moderate CE and to compare these effects with those of RE on cognitive function, utilizing behavioral measures as the primary assessment tool. Based on previous findings (*Chang et al., 2017*; *Wen & Tsai, 2020*), this study expected that the behavioral performance measures would be higher after the two exercise sessions compared to a control session of sitting.

Furthermore, this study hypothesized that behavioral performance after CE would be better than that after RE in both response time and accuracy rate.

## METHODS

### Participants and ethics statement

This study recruited 15 males between the ages of 21 and 34 years from the University of Taipei. The eligible participants were healthy but without a habit of regular exercise in the previous three months. Additional selection criteria were (a) being free from any chronic disease, *i.e.,* cardiovascular disease and pulmonary disorders; (b) no color blindness, color weakness, or any other indication of abnormal vision; (c) no neurological disorder or any psychiatric illness; and (d) no to all of items in the Physical Activity Readiness Questionnaire (PAR-Q) to ensure the safety of participants during the exercise intervention. Informed consent was obtained from all participants after they agreed to enroll in this study, which was approved by the Institutional Review Board of the University of Taipei, Taipei City, Taiwan (IRB-2020-010). The study also complied with the latest version of the Helsinki Declaration.

### Experimental procedure

The study consisted of one screening visit and three experimental sessions at the weight training room of the Department of Aquatic Sport, University of Taipei (Tian-Mu campus) from November 2020 to June 2021. These sessions were conducted over approximately 4 weeks with a week-long rest interval and counterbalanced order. Participants were instructed to avoid any strenuous exercise for 24 h and abstain from ergogenic aids, alcohol, caffeine, and medications before each experimental session to rule out possible effects on exercise and cognitive performance. The test environment was set at a minimum noise level and a temperature of 24–26 °C. To control alterations to the circadian rhythm, all participants engaged in the exercise sessions in the morning (8 am to 12 pm).

On their first visit to the laboratory, the participants received a detailed explanation of each experimental procedure. Before each formal intervention began, all participants underwent a pre-experiment assessment, including $\dot{V}O_{2peak}$ and 10-repetition maximum (10 RM) assessment of the upper extremity, in the laboratory. Their body composition was also measured using bioelectrical impedance analysis. In addition, participants were instructed to practice and familiarize themselves with the cognitive task. All participants were instructed to complete practice trials of the Stroop Color and Word Test for all three conditions until they achieved an accuracy rate of 85% (*Chang et al., 2015*).

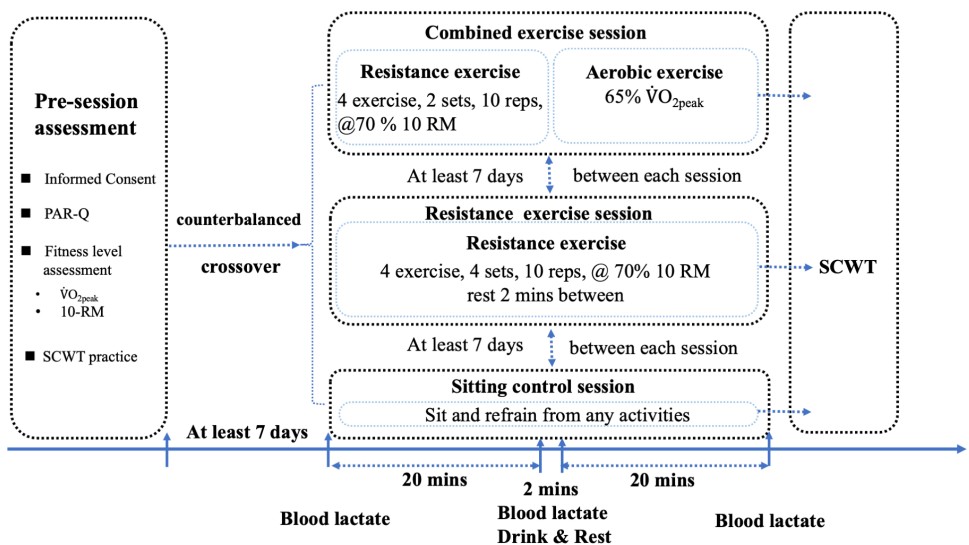

**Figure 1** **Experimental protocol.** PAR-Q, physical activity readiness questionnaire. 10 RM, 10 repetition maximum. SCWT, Stroop color world test. The order of experimental sessions for participants was randomized and counterbalanced.

The experimental sessions comprised CE, RE, and sitting control (SC) sessions. The heart rate (HR), blood lactate, and rating of perceived exertion (RPE) were used to assess the exercise manipulation. To measure the HR, a telemetric BioHarness™ 3.0 chest strap (Zephyr™ Technology, Medtronic, Annapolis, USA) was secured round the participant's chest just below the sternum. It detected and recorded the HR continually and shared it *via* Bluetooth (802.15.4 protocol) with the Zephyr™ Echo Gateway. The HR reading was displayed on a laptop monitor throughout each intervention session. After finishing each session, the heartbeat data were exported *via* OmniSense™ Live Software version 5.0. Blood lactate levels (mmol/L) were assessed at three time points: pre-session, during the session (approximately 20 min after the initiation of each session), and post-session. The measurements were conducted using a lactate analyzer (EDGE: Blood Lactate Monitoring System, Hsinchu Science Park, Hsinchu, Taiwan, ROC). In the case of RPE, the Borg psychophysiological rating (6–20 scale) (*Borg, 1970*; *Borg, 1982*) was applied to assess individuals' subjective psychophysiological feelings regarding their RPE. This study also used the RPE Borg 6–20 scale to record participants' 10 RM RE and aerobic exercise assessment exertion status. Furthermore, RPE was recorded at five-minute intervals during each RE and CE session. The experimental procedure is shown as Fig. 1.

## Cardiorespiratory fitness and resistance exercise assessment

The cardiorespiratory fitness and resistance exercise assessment was used to determine the exercise intensity setting for the CE and RE sessions. To assess the cardiorespiratory fitness level, a bicycle ergometer (Lode Corival 906900, Lode Medical, Groningen, the Netherlands) was used. Before assessment, all participants were measured for their body weight and body composition *via* bioelectrical impedance analysis (InBody 270, Cerritos,

CA, USA). Thereafter, all participants were asked to rest for five minutes; they sat on the bicycle ergometer while the experiment operator recorded each participant's seat height and the flexion angles of the knee joint to ensure that the participants could ride the bicycle ergometer at a fixed seat height and trunk angle during the interventions. The participants started to cycle, with the first four minutes being the warm-up phase and the workload kept at 45 W; after four minutes, the graded exercise testing (GXT) protocol increased the workload by 15 W every minute. A certified exercise physiologist monitored the participants' cycling revolutions per minute (RPM) to ensure that the RPM interval was between 75 and 85 RPM to maintain peddle speed. During the assessment, the experiment operator and certified exercise physiologist kept encouraging the participants persist until the RPE was above 18 and the participants failed to continue the assessment or the RPM was below 60 (*Riechman et al., 2002*). In addition, the HR and RPE were recorded every half minute of each stage. At the end of the assessment, the experiment operator recorded the ending stage workload and calculated $\dot{V}O_{2peak}$ using the following formula for males:

$$\dot{V}O_{2peak} = 10.51 * (\text{watts at ending stage}) + 6.35 * (\text{body weight})$$
$$- 10.49 * (\text{participant's age}) + 519.3 \text{ml/min}$$

(*Storer, Davis & Caiozzo, 1990*).

In the case of RE, the 10 RM of each exercise was measured. This experiment applied four resistance training exercises: supine Smith machine bench press; Smith machine bent-over row; standing dumbbell lateral raise; and seated dumbbell biceps curl. These exercises involved five major muscle groups: chest, latissimus, deltoids, triceps, and biceps. The RE assessment involved several steps. In the beginning, all participants had to first finish the standard warm-up procedure: running for 150 m and dynamic warm-ups, including forward lunge and trunk rotation, reverse lunge and trunk rotation, knee hug and heel raise, single leg stance, and contralateral side arm raise. After the dynamic warm-ups, all participants began to perform the 10 RM assessment. Before the assessment started, a certified strength and conditioning specialist asked the participants whether or not they understood or were familiar with the exercise movement. If a participant did not have prior weight training experience, the certified strength and conditioning specialist demonstrated the correct form and technique of the exercise and confirmed that the participants were in good form to continue the assessment. In the compound movements, including supine Smith machine bench press and Smith machine bent-over row, the participants started with the empty bar for warm-up and continued with 50% of their body weight; for single joint movements, such as standing dumbbell lateral raise and seated dumbbell biceps curl, participants started with 2-kg dumbbells for warm-up and continued with 10% of their body weight. Whether or not the participants were novices, the strength assessment increased the weight gradually according to the RPE and participants' subjective feedback to determine the final 10 RM (*Suchomel et al., 2021*).

## Sessions

Upon arrival, all participants were instructed to practice the cognitive task before wearing the HR strap. After the first 20 min of each session were counted, all participants were asked

to rest for two minutes and drink 150 ml of water. During these sessions, the participant was supervised by an American College of Sport Medicine (ACSM)-certified exercise physiologist to ensure appropriate exercise intensity manipulation and safety.

### Combined exercise session

The CE session included 10 repetitions of two sets of an upper body RE consisting of Smith machine bench press, Smith machine bent-over row, standing lateral dumbbell raise, and seated dumbbell curl. The participants were instructed to finish the two sets of the first RE and then continue the second exercise in the aforementioned order and so on. The rest period between each set lasted two minutes. The participants finished the CE with 65% $\dot{V}O_{2peak}$ ergometer cycling.

### Resistance exercise session

In the RE session, the participants had to perform four sets of upper body exercises, including Smith machine bench press, Smith machine bent-over row, standing dumbbell lateral raise, and seated dumbbell curl. Participants finished the exercises in the aforementioned order and performed 10 repetitions of each set and rested for two minutes between sets.

### Sitting control session

The sequence of the SC session was the same as that of the aforementioned sessions, except that all participants were instructed to sit and rest quietly for 40 min in a non-distraction room, refrain from sleeping, reading, talking, writing, or using their cellphone.

## Stroop Color and Word Test

This study administered the Stroop Color and Word Test (SCWT), which is extensively applied to determine manifold cognitive functions, especially the executive function (*Chang et al., 2016*; *Scarpina & Tagini, 2017*). The current study employed the paper version of the SCWT, which has been widely utilized in clinical studies (*Caffarra et al., 2002*; *Valgimigli et al., 2010*). The response time was measured using a calibrated digital stopwatch (CASIO: HS-80TW-1) and was also independently measured by the same laboratory assistant to ensure inter-rater reliability. The SCWT is with three conditions, *i.e.,* neutral, congruent, and incongruent. Each condition contained 50 stimulus words in random order (10 words in one column and five words in one row) on A4 paper. In the neutral condition, the stimulus words were in black and were Chinese characters for the names of three colors, *i.e.,* red, green, and blue. In the congruent condition, the three color names were presented in their colors. Last, in the incongruent condition, the three color names were not presented in their colors, *e.g.,* the Chinese character for red might be in blue or green color, and the participants had to state the color of the Chinese character instead of the color name. All participants had to read the Chinese characters aloud and clearly for each condition, and as fast as possible in the correct executive pattern mentioned above. The response time(s) was the time taken to complete each condition. The accuracy rate was calculated as the number of false responses to stimuli divided by the total number of stimuli and given as a percentage.

**Table 1  Participant demographic and anthropometric characteristic, mean (S.D.).**

| Characteristics | Participants (15 males) Mean (S.D.) |
| --- | --- |
| Age (years) | 24.73 (3.49) |
| Height (cm) | 173.23 (6.51) |
| Weight (kg) | 73.69 (10.93) |
| Body Mass Index (kg/m$^2$) | 24.47 (2.61) |
| Muscle mass (kg) | 33.69 (6.09) |
| Body fat mass (kg) | 14.43 (5.62) |
| Body fat percentage (%) | 19.50 (6.64) |
| $\dot{V}O_{2peak}$ (ml kg$^{-1}$ min$^{-1}$) | 44.13 (7.06) |

**Notes.**
S.D., standard deviation.

## Statistical analysis

The statistical analyses were performed with SPSS *v.* 23 for Mac OS (Chicago, IL, USA). Descriptive statistics (means and standard deviations) are used to present demographic and anthropometric characteristics and exercise manipulation data. A 3 (session: CE, RE, and SC) × 3 (time point: pre-session, during session, and post-session) repeated-measures analysis of variance (ANOVA) was performed for HR and blood lactate level to assess the effectiveness of the exercise manipulation. For behavioral data, a 3 (session: CE, RE, and SC) × 3 (Stroop condition: neutral, congruent, and incongruent) repeated-measures ANOVA was performed separately for the response time and accuracy rate. The response time and accuracy rate were analyzed after the Greenhouse–Geisser epsilon correction was made when the assumption of sphericity was not met. Post-hoc comparisons were made using Bonferroni-adjusted multiple comparisons. Effect sizes are reported using partial eta squared ($\eta_p^2$). Means and standard deviations are presented. The alpha level was set at 0.05.

## RESULTS

Descriptive statistics for participant demographic and anthropometric characteristics and cardiovascular fitness assessment results were shown in Table 1. Descriptive data for the RE and AE exercise intensity manipulation were shown in Table 2.

### Intervention manipulation assessment
*Lactate*

The two-way ANOVA revealed a significant main effects of intervention ($F(2, 28) = 127.08$, $p < .001$, $\eta_p^2 = .90$), with a higher mean lactate for the CE ($4.60 \pm 0.18$ mmol/L, $p < .001$) and RE ($4.09 \pm 0.23$ mmol/L, $p < .001$) than for the SC ($1.36 \pm 0.09$ mmol/L). However, the mean lactate was not significantly different between the RE and CE ($p = .14$). The main effect of time was significant ($F(2, 28) = 140.84$, $p < .001$, $\eta_p^2 = .91$), with a higher mean lactate in the during-intervention ($4.14 \pm 0.17$ mmol/L, $p < .001$) and post-intervention ($4.53 \pm 0.23$ mmol/L, $p < .001$) than in the pre-intervention ($1.37 \pm .06$ mmol/L).

**Table 2  Descriptive data for the resistance exercise & aerobic exercise intensity manipulation.**

| Resistance exercise movements | 10 RM Mean (S.D.) |
| --- | --- |
| Smith machine bench press | 49.50 (18.18) |
| Smith machine bent-over row | 53.67 (17.06) |
| Standing dumbbell lateral raise | 8.87 (2.86) |
| Seated dumbbell biceps curl | 10.00 (3.12) |
| Aerobic exercise intensity | Watts |
| 65% $\dot{V}O_{2peak}$ | 144.42 (34.39) |

**Notes.**

10 RM, 10 repetition maximum; S.D., standard deviation.

However, the mean lactate was not significantly different between the during-intervention and post-intervention ($p = .25$).

A significant intervention × time interaction was revealed ($F(4, 56) = 51.76, p < .001, \eta_p^2 = .79$). Regarding simple effects of intervention, the pre-intervention mean lactate was not significantly different among three interventions (CE: $1.40 \pm 0.12$ mmol/L; RE: $1.31 \pm 0.09$ mmol/L; and SC: $1.40 \pm 0.16$ mmol/L, $ps = 1.00$). During the intervention, both CE ($5.19 \pm 0.35$ mmol/L, $p < .001$) and RE ($5.81 \pm 0.33$ mmol/L, $p < .001$) induced higher mean lactate than SC ($1.43 \pm 0.13$ mmol/L), and there was no significant difference between CE and RE ($p = .65$). After the intervention, the mean lactate was highest in CE ($7.19 \pm 0.35$ mmol/L), followed by RE ($5.15 \pm 0.44$ mmol/L), and SC ($1.25 \pm 0.09$ mmol/L; $ps < .01$). Regarding simple effects of time, in CE session, the mean lactate was highest in post-intervention, followed by during-intervention, and pre-intervention ($ps < .01$). In the RE session, the mean lactate of post-intervention ($p < .001$) and during-intervention ($p < .001$) was higher than SC, and there was no significant difference between post-intervention and during intervention ($p = .24$). In the SC session, there was no significant difference among three time of intervention ($ps > .05$).

### Heart rate

The two-way ANOVA revealed a significant main effect of intervention ($F(1.37, 19.16) = 114.57, p < .001, \eta_p^2 = .89$), the whole session mean heart rate was highest in CE (CE: $106.16 \pm 2.81$ bpm), followed by RE (RE: $82.88 \pm 2.63$ bpm), and SC ($69.79 \pm 1.79$ bpm; $ps < .001$). The main effect of time was significant ($F(2, 28) = 84.89, p < .001, \eta_p^2 = .86$), with a higher mean heart rate in the during-intervention ($95.91 \pm 1.85$ bpm, $p < .001$) and post-intervention ($93.22 \pm 2.95$ bpm, $p < .001$) than in the pre-intervention ($69.69 \pm 2.20$ bpm). However, there was no significant difference between during-intervention and post-intervention ($p = .78$).

A significant intervention × time interaction was revealed ($F(4, 56) = 34.48, p < .001, \eta_p^2 = .71$). Regarding simple effects of intervention, the pre-intervention mean heart rate did not differ significantly among three interventions (CE: $71.80 \pm 3.43$ bpm; RE: $66.73 \pm 2.65$ bpm; and SC: $70.53 \pm 2.85$ bpm; $ps > .05$). During the intervention, the mean heart rate was highest in CE ($125.35 \pm 3.90$ bpm), followed by RE ($92.23 \pm 2.00$ bpm), and SC ($70.16 \pm 2.21$ bpm, $ps < .001$). After the intervention, the mean heart rate was

highest in CE (121.33 ± 4.05 bpm), followed by RE (89.67 ± 4.58 bpm), and SC (68.67 ± 2.69 bpm; $ps < .01$). Regarding simple effects of time, in the CE session, the mean heart rate of during-intervention ($p < .001$) and post-intervention ($p < .001$) were significantly higher than pre-intervention. However, there was no significant difference between post-intervention and during-intervention ($p = .80$). In the RE session, the mean heart rate of post-intervention ($p < .001$) and during-intervention ($p < .001$) were higher than pre-intervention. However, there was no significant difference between during-intervention and post-intervention ($p = 1.00$). In the SC session, there was no significant difference among three time of intervention ($ps = 1.00$).

### RPE

The two-way ANOVA revealed a significant main effect of intervention ($F(2, 28) = 85.52$, $p < .001$, $\eta_p^2 = .86$), with higher mean RPE in CE (10.05 ± 0.29, $p < .001$) and RE (9.28 ± 0.31, $p < .001$) than in SC (6.29 ± 0.08). However, there was no significant difference between CE and RE ($p = .13$). The main effect of time was significant ($F(2, 28) = 116.52$, $p < .001$, $\eta_p^2 = .89$), with a highest mean RPE in the during-intervention (10.66 ± 0.24), followed by post-intervention (8.67 ± 0.33), and pre-intervention (6.29 ± 0.10, $ps < .001$).

A significant intervention × time interaction was revealed ($F(4, 56) = 43.75$, $p < .001$, $\eta_p^2 = .76$). Regarding simple effects of intervention, the pre-intervention mean RPE was not significantly different among three interventions (CE: 6.20 ± 0.11; RE: 6.40 ± 0.19; SC: 6.27 ± 0.15; $ps = 1.00$). During the intervention, the mean RPE of CE (13.14 ± 0.35, $p < .001$) and RE (12.56 ± 0.45, $p < .001$) were higher than SC (6.27 ± 0.12), and there was no significant difference between the CE and RE ($p = .34$). After the intervention, the mean RPE of CE (10.80 ± 0.60, $p < .001$) and RE (8.87 ± 0.60, $p < .001$) were higher than SC (6.33 ± 0.13), and there was no significant difference between the CE and RE ($p = .08$). Regarding simple effects of time, in CE session, the mean RPE of during-intervention was highest, followed by post-intervention, and pre-intervention ($ps < .01$). In the RE session, the mean RPE of during-intervention was highest, followed by post-intervention, and pre-intervention ($ps < .01$). In the SC session, there was no significant difference among three time of intervention ($ps = 1.00$).

All results of physiological variables were presented in Table 3.

## Behavioral data
### Response time

The main effect of intervention on response time was significant ($F(2, 28) = 3.91$, $p = .03$, $\eta_p^2 = .22$), with a shorter mean response time for the CE (21.49 ± 1.04 s) than for the SC (23.72 ± 1.00 s; $p < .05$). Furthermore, the mean response time of RE (22.03 ± 1.13 s) was not significantly different with CE ($p = 1.00$) and SC ($p = .23$). The main effects of condition was significant ($F(1.07, 15.01) = 97.00$, $p < .001$, $\eta_p^2 = .87$), with a shorter mean response time for the neutral (17.32 ± 0.59 s) and the congruent conditions (17.01 ± 0.66 s) than for the incongruent condition (32.92 ± 1.89 s). However, there was no significant difference between the neutral and the congruent conditions. The intervention × condition interaction was not significant ($F(2.47, 34.52) = 1.20$, $p = .32$, $\eta_p^2 = .079$; see Fig. 2A).

**Table 3  Descriptive data for exercise manipulation check, mean (S.D.).**

| Session | Variables | | |
|---|---|---|---|
| | Lactate (mmol/L) | Heart Rate (bpm) | RPE |
| **CE** | | | |
| Pre exercise | 1.40 (0.46) | 71.80 (13.27) | 6.20 (0.41) |
| During exercise | 5.19 (1.35) | 125.35 (15.11) | 13.14 (1.36) |
| Post-exercise | 7.19 (1.37) | 121.33 (15.67) | 10.80 (2.34) |
| **RE** | | | |
| Pre exercise | 1.31 (0.35) | 66.73 (10.26) | 6.40 (0.74) |
| During exercise | 5.81 (1.27) | 97.23 (7.75) | 12.56 (1.74) |
| Post-exercise | 5.15 (1.69) | 89.67 (17.75) | 8.87 (2.33) |
| **SC** | | | |
| Pre exercise | 1.40 (0.62) | 70.53 (11.03) | 6.27 (0.59) |
| During exercise | 1.43 (0.51) | 70.16 (8.55) | 6.27 (0.46) |
| Post-exercise | 1.25 (0.33) | 68.67 (10.40) | 6.33 (0.49) |

Notes.

CE, combined exercise; RE, resistance exercise; SC, sitting control; S.D., standard deviation.

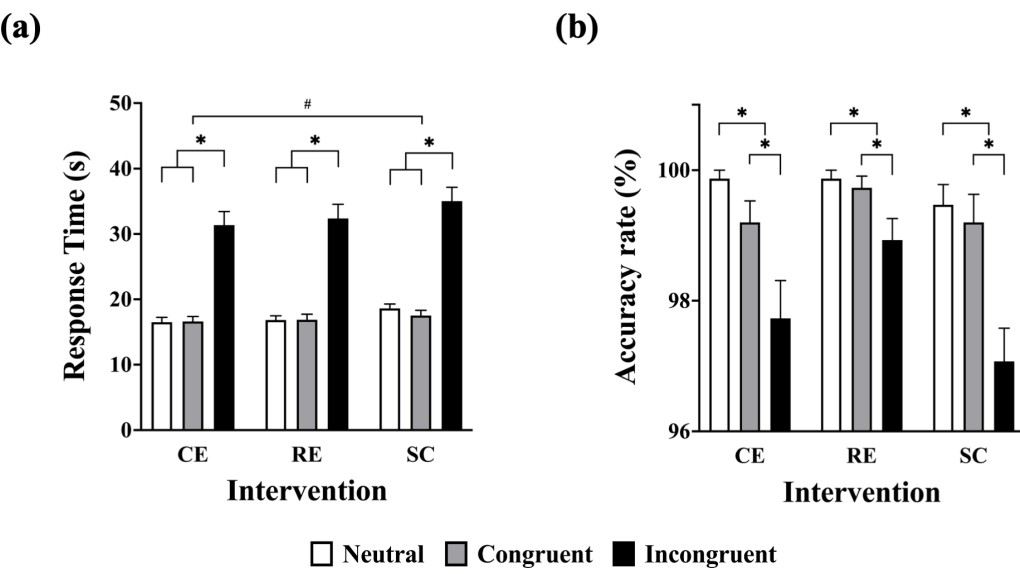

**Figure 2  Response time and accuracy rate among sessions in the Stroop Color and Word Test.** CE, concurrent exercise session; RE, resistance exercise session; SC, sitting control session; *, significant difference between Stroop conditions; #, significant difference between the CE and the SC.

### Accuracy rate

The main effect of intervention on the accuracy rate was not significant ($F(2, 28) = 3.23$, $p = .06$, $\eta_p^2 = .19$; CE: $98.93 \pm 0.28\%$; RE: $99.51 \pm 0.15\%$; SC: $98.58 \pm 0.30\%$). The main effects of condition was significant ($F(1.27, 17.71) = 32.63$, $p < .001$, $\eta_p^2 = .70$), with a highest accuracy rate for the neutral ($99.73 \pm 0.13\%$), followed by congruent ($99.38$

$\pm$ 0.15%), and incongruent conditions (97.91 $\pm$ 0.27%; $ps < .05$). The intervention $\times$ condition interaction was not significant ($F(4, 56) = 1.81$, $p = .14$, $\eta_p^2 = .12$.; see Fig. 2B).

## DISCUSSION

The present study explored whether acute CE affects cognitive function. To the best of our knowledge, this study is among the first to investigate the effects of acute CE on cognitive function in sedentary males and compare the results to the effects of traditional sequential movement resistance training. Our main findings demonstrated that acute CE, with 20 min of upper extremity RE, combined with 20 min of lower extremity cycling on an ergometer, decreased the SCWT response time, regardless of the SCWT conditions, *i.e.,* neutral, congruent, and incongruent, and the effects were similar to RE. Furthermore, accuracy rates showed nonsignificant differences between the three sessions. While these findings are consistent with previous research demonstrating that a single bout of CE, consisting of low-intensity resistance and aerobic exercise, may enhance the SCWT response time in a young female college student population (*Chang et al., 2017*), our findings extend the previous study by demonstrating cognitive benefits after moderate-intensity CE consisting of upper extremity RE and lower extremity AE. Our study thus highlights potential avenues of exploration for prescribing CE, in relation to intensity, duration, type, order of exercise, and applied population domain, to induce optimal cognitive performance.

Our findings pertaining to the different cognitive conditions in the SCWT are consistent with previous research (*Chang et al., 2015*; *Chang et al., 2014*; *Wang et al., 2019*). These studies have consistently observed the presence of the ''Stroop effect'', wherein response times were slower and accuracy was reduced in the incongruent condition compared to the congruent condition. The alignment of our results with these prior findings lends support to the validity of our manipulation of the cognitive task in the present study. Additionally, the observed improvements in cognitive performance in both RE and CE were in line with a previous meta-analysis indicating that acute RE leads to moderate cognitive improvements compared to a no-exercise control (*Wilke et al., 2019*). A randomized controlled trial with a small sample size showed that acute CE leads to better SCWT performance, compared to high-intensity RE and AE (*Chang et al., 2017*). However, our findings are inconsistent with those of a study showing that acute CE does not enhance the behavioral indices of SCWT (*Wen & Tsai, 2020*). However, the acute CE in the previous study consisted of 30 min of moderate-intensity body weight or dumbbell RE (12–16 repetitions) combined with a moderate-intensity aerobic dance exercise (55% of heart rate reserve (HRR)) in circuit-training form (*Wen & Tsai, 2020*); therefore, it is possible that both the RE and AE might not have been intense enough to improve behavioral performance in the SCWT (*Braga et al., 2022*; *Peruyero et al., 2017*; *Zhou & Qin, 2019*)

In addition, the duration of exercise (*Basso & Suzuki, 2017*; *Chang et al., 2019*) as well as the type of exercise (*Fiorelli et al., 2019*; *Hacker et al., 2020*; *Ishihara et al., 2021*) may exert an influence on the acute effects of exercise on domain-specific cognitive function, potentially varying across different populations, age groups, and genders. *Chang et al. (2015)* found that the acute exercise duration has dose–response relationships with

acute-exercise-induced cognitive function improvement. They suggested that 20 min of acute moderate-to-vigorous intensity exercise might be the optimal modality for basic information processing and executive function improvements. Thus, the findings of the present study corresponded with the results obtained by the aforementioned studies, since the RE was composed of four RE movements with four sets and 10 repetitions, with rest periods of two minutes between sets, *i.e.,* the actual exercise duration of the RE was approximately 15 min. Compared to the RE, the CE was composed of RE and AE in a single bout of the session; as a result, the total exercise duration of the CE might be too lengthy to induce better executive function performance due to physiological and mental fatigue, which might be the reason for the lack of significant differences between CE and RE regimens in response times and accuracy rates of the SCWT.

Different types of CE modes might also affect the results. *Wen & Tsai (2020)* showed that acute CE, consisting of a combination of aerobic dance as an AE and circuit training involving body weight and dumbbell RE, might improve neurophysiological inhibition control but not the performance of behavioral indices among obese women. *Chang et al. (2017)* specified another mode of CE combining machine-based circuit-training RE (30% 1 RM, 12 repetitions) with low-intensity walking on a treadmill (50–60% of HRR), which may enhance both the accuracy rate and response times in the SCWT. In contrast, the present study did not observe a significant difference in accuracy rate between the three sessions. As a result, irrespective of exercise duration and intensity, the exercise type might also play a significant mediator role in exercise-induced enhancement of cognitive function.

To date, only a few experimental studies have explored the mechanisms behind improved cognitive function after acute CE. The general exercise-induced improvements in cognition might be influenced by blood or systemic lactate levels (*Ben-Zeev, Shoenfeld & Hoffman, 2022*). Lactate is not only an important energy source for replacing glucose during exercise, but also a signaling molecule for modulation of neural plasticity through specific activation of signal receptors and subsequent downstream signaling pathways (*Xue et al., 2022*). The degree of exercise-induced executive function improvements may be associated with blood lactate levels, which is probably due to elevated cerebral lactate metabolism, since prolonged exercise may decrease cerebral oxygenation. Furthermore, lactate, as an exercise-induced myokine, may play an important role as mediator and modulator of brain adaptation and be a myokine for activating certain brain areas (*Chen et al., 2021*; *Hashimoto et al., 2021*; *Severinsen & Pedersen, 2020*). In the present study, blood lactate levels were not altered significantly during SC, but they were slightly different and increased during RE and CE. *Huang et al. (2021)* found that moderate-intensity RE increased cerebral lactate metabolism immediately after exercise, compared to pre-exercise. As a result, the potential impact of CE and RE on postexercise EF enhancements may be due to the increase in blood lactate levels, the metabolism of cerebral lactate, and cerebral oxygenation. *Chang et al. (2017)* also indicated a similar potential mechanism (*via* the tissue oxygen index) in the prefrontal cortex, implying that low-intensity CE enhances oxygen and nutrient distribution throughout the brain *via* oxygen saturation and increased blood flow and that

increased oxygenated hemoglobin concentration in the prefrontal cortex might activate cognition-related nerves, consequently improving exercise-induced cognitive performance.

In addition to the manipulation of exercise intervention parameters such as intensity, duration, and type of exercise, the enhancements in EF induced by CE may also be influenced by other factors, including variations in lactate levels and cerebral oxygenation levels. Apart from the aforementioned mechanism potentially influencing exercise-induced EF improvements, recent studies have also investigated the relationship between exercise-induced EF enhancement and arousal based on arousal–performance interaction theory (*Chou et al., 2021*). Acute bouts of RE might induce alterations of the endocrine system, nervous system, and subsequent changes in the level of arousal (*Tsukamoto et al., 2017*). *Wang et al. (2019)* also suggested that exercise may induce eustress, which may be correlated with EF improvement and the neural activation process *via* neuroendocrinological responses. Their study showed that exercise-induced modification of the perceptual response may also be associated with the locus coeruleus-norepinephrine (LC-NE) functional coupling system, which may also modulate acute exercise-induced cognitive performance (*Grueschow, Kleim & Ruff, 2022*). Collectively, despite the lack of a direct evidential relationship between exercise-induced cognitive performance and phasic activation of the LE-NE system, the aforementioned research contributes to a broader understanding of the potential factors and mechanisms that may underlie the manifestation of improved performance on the SCWT following CE.

The main strength of the present study is the investigation of the effects of acute CE on multiple aspects of executive function in sedentary male individuals. However, the study also has some limitations. Firstly, acute exercise interventions might be influenced by participants' daily lifestyle behaviors, physical activity status, and dietary habits outside of the intervention period. While our study employed a counter-balanced design, similar to previous studies (*Drollette et al., 2014*; *Hillman et al., 2009*; *Ludyga et al., 2020*), it is important to acknowledge that day-to-day variability could not be completely eliminated as a potential confounding factor. Despite reminding and instructing all participants to maintain their regular daily lifestyles and levels of physical activity, it is possible that some individuals did not strictly adhere to these guidelines. Future studies should consider incorporating pre-test and post-test measurements within a crossover experimental design to mitigate the impact of such variability. In addition, the duration of the exercise interventions in this study might have been influenced by exercise type, *i.e.,* the actual exercise duration of RE was less than that of other sessions due to the rest periods between sets. Given the exploratory nature of our study, we recruited as many participants as possible, resulting in a sample size of $n = 15$, which is consistent with the experimental design employed in previous similar studies (*Chang et al., 2014*; *Tsukamoto et al., 2017*). It is important to note that the relatively small sample size and the inclusion of only sedentary male adults in our study may limit the generalizability of our findings to the broader population. Future studies should aim to recruit larger sample sizes to enhance statistical power and improve generalizability. Furthermore, since we conducted only one cognitive assessment following each session, the duration of the session-induced increase in cognitive task performance remains unclear. Further research on CE should be undertaken

with different orders, intensities, patterns, and exercise durations in order to investigate the optimal combination that might induce the highest enhancement of cognitive function performance.

## CONCLUSION

This study demonstrated that a single session of moderate-intensity CE resulted in a significant improvement in response time in the SCWT. The effect of CE was found to be comparable to that of RE. These findings highlight the potential effectiveness of CE as a viable exercise modality for enhancing cognitive function. Considering the recognized efficacy and accessibility of CE, as well as its additive physiological benefits observed across different settings, future research should continue to explore and deepen our understanding of the cognitive benefits associated with this type of exercise, along with other exercise modalities.

### Funding

This work was supported by the National Science and Technology Council, Taiwan (MOST109-2410-H845-014). The funders had no role in study design, data collection and analysis, decision to publish, or preparation of the manuscript.

### Grant Disclosures

The following grant information was disclosed by the authors:
National Science and Technology Council, Taiwan: MOST109-2410-H845-014.

### Competing Interests

Yu-Kai Chang is an Academic Editor for PeerJ. The other authors declare that they have no competing interests.

### Author Contributions

- Ying-Chu Chen conceived and designed the experiments, performed the experiments, prepared figures and/or tables, and approved the final draft.
- Ruei-Hong Li performed the experiments, analyzed the data, prepared figures and/or tables, and approved the final draft.
- Feng-Tzu Chen performed the experiments, analyzed the data, prepared figures and/or tables, and approved the final draft.
- Chih-Han Wu performed the experiments, analyzed the data, prepared figures and/or tables, and approved the final draft.
- Chung-Yu Chen conceived and designed the experiments, analyzed the data, authored or reviewed drafts of the article, and approved the final draft.
- Che-Chien Chang conceived and designed the experiments, analyzed the data, authored or reviewed drafts of the article, and approved the final draft.
- Yu-Kai Chang conceived and designed the experiments, analyzed the data, authored or reviewed drafts of the article, and approved the final draft.
## Human Ethics

The following information was supplied relating to ethical approvals (*i.e.*, approving body and any reference numbers):

All the authors affirm that this work complies with the ethical standards approved by the Institutional Review Board at the University of Taipei, Taipei City, Taiwan (IRB-2020-010) in accord with the 2013 Declaration of Helsinki. Informed consent was obtained from all participants included in the study.

## Data Availability

The raw data are available in the Supplemental File.

## Supplemental Information

Supplemental information for this article can be found online at http://dx.doi.org/10.7717/peerj.15768#supplemental-information.

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
