# Peer review of "Acute effect of combined exercise with aerobic and resistance exercises on executive function"

_PeerJ, doi:10.7717/peerj.15768_

## Round 0.1 · original submission · Major Revisions

Both Reviewers have raised important questions and provided useful suggestions to improve the clarity of the manuscript. I would add a minor comment that the figures should be in order of appearance in the text.

Reviewer 1 ·

Basic reporting

The manuscript presents a well-designed study that explores the effects of combined aerobic and resistance exercise on cognitive function. In my opinion, the article provides significant scientific and practical value for readers of PeerJ. The manuscript is well-written, with a clear and consistent structure that makes it easy to follow. I would like to congratulate the authors on their well-prepared experimental work.

Experimental design

However, I would like to draw attention to a few minor issues that require clarification. First, the experimental protocol needs further clarification, particularly regarding the adaptation period to the Stroop Color and Word Test before the testing session. Additionally, it would be helpful to specify the exact minutes of lactate sampling during the testing procedure (lines 132-133).

Furthermore, the Borg psychophysiological rating was used to assess the subjective psychophysiological feelings of individuals regarding their RPE (lines 135-138). However, the results of this measurement were not presented in the manuscript. Moreover, the study recruited only 15 male participants, and it would be helpful to know whether this sample size was sufficient to achieve adequate test power and whether an a priori analysis was conducted (line 90).

Validity of the findings

The findings of the study appear valid and provide significant scientific and practical value for readers of PeerJ. However, in the limitations section of the discussion, it would be appropriate to expand on the effects of gender and the lack of generalizability to the wider population. Furthermore, in the results analysis, it would be beneficial to indicate the effect size for post-hoc tests.

Additional comments

The abstract should be more specific, particularly in the results section. It is recommended to include the most important statistics to provide more detail.

Reviewer 2 ·

Basic reporting

This study examined the effects of combined exercise affects executive function. The authors indicated that combined exercise leads to beneficial effects on cognitive function as compared with sitting control condition. After reading the manuscript, I have some comments and concerns on this manuscript, including experimental design. Hopefully, my comments below are useful to the authors for their study.

Experimental design

First, I am wondering why the authors assessed executive function only after exercise/control. Without pre-exercise data, the effects of exercise are obscure.
Second, why did the authors compare combined exercise and resistance exercise. Rationale behind the comparison between these two conditions are still unclear in the Introduction.

Validity of the findings

The present results indicate that effects of combined exercise are comparable among neutral, congruent, and incongruent conditions. What do the authors interpret these results in regards to cognitive demands? Since these results are main findings, more discussion is necessary.

This study indicated that combined exercise improved cognitive performance, while resistance exercise did not. However, blood lactate level increased in a similar manner in both exercise conditions. If blood lactate plays a role in improvements, the lack of improvement in resistance exercise is difficult to interpret.

Additional comments

Title: it may be better to add what was combined, if possible.
Is paper version of Stroop Color and Word Test accurate? How was time measured?
In my opinion, results of physiological variables are easy to follow in a table.

---

## Round 0.2 · accepted · Accept

The authors have addressed all comments, concerns, and suggestions from the reviewers.

Reviewer 1 ·

Basic reporting

no comment

Experimental design

no comment

Validity of the findings

no comment

Additional comments

The authors have addressed my comments.I have no further comments to add.